# To Use or not to Use Muon: How Simplicity Bias in Optimizers Matters

**Sara Dragutinović, Rajesh Ranganath**
Courant Institute School of Mathematics, Computing, and Data Science, NYU
`sara.dragutinovic@nyu.edu`

## Abstract

Out of the recently introduced optimizers, Muon has perhaps gained the highest popularity due to its superior training speed. While many papers focus on the benefits of Muon, our paper questions if there are any downsides this speedup brings. We explore the biases induced when optimizing with Muon, providing theoretical analysis and its consequences to the learning trajectories and solutions learned. While the theory does explain the benefits Muon brings, it also guides our intuition when coming up with a couple of examples where Muon-optimized models may be disadvantaged, due to losing a simplicity bias. More broadly, this paper should serve as a reminder: when developing new optimizers, it is essential to consider the biases they introduce, as these biases can fundamentally change a model's behavior—for better or for worse.

## 1 Introduction

Employing deep learning in practice usually involves careful decisions on the model architecture and data (e.g. which augmentations or feature engineering), while the choice of the optimizer typically relies on established defaults such as Adam, or more recently, Muon. This paper aims to spotlight the impact of optimizer choice: different optimizers take different optimization paths in the loss landscape, each with their own inductive biases. Despite this, prior works primarily focus on the benefit of faster optimization (Wen et al., 2026). There remains a lack of theory or intuition on how the learning trajectory taken impacts functional properties of the learned solution.

Specifically, in this work we focus on the Muon optimizer Jordan et al. (2024b), which seems to have attracted a large audience in quite a short period of time. Its performance on NanoGPT Speedrun competition Jordan et al. (2024a) has been largely responsible, as Muon converges to a certain validation loss much quicker (in wallclock time) than other, older and newer optimizers. People seem to have adopted Muon as one of the new defaults: it is the default optimizer for training NanoChat Karpathy (2025), and perhaps in industry as well, as hypothesized by Su (2025). But we still don't know Muon's biases, we don't know which trajectory in the loss landscape Muon takes to be so quick, and we don't know implications of that to the solution Muon-optimized model converges to!

Our goal in this paper, apart from emphasizing asking those questions, is to make initial steps in answering them. We set to do so by analyzing Muon's behavior in deep linear networks, and theoretically explaining how optimizing with Muon (via Spectral GD) removes a simplicity bias that gradient descent (GD) exhibits Nakkiran et al. (2019); Gidel et al. (2019); Refinetti et al. (2023); Zhang et al. (2025a). This theory explains the speed-up benefits of Muon, e.g. in imbalanced modalities setting, as observed by Vasudeva et al. (2025); Wang et al. (2025). However, the downside of the speed-up is losing the simplicity bias. Guided by this loss in simplicity, we develop a few settings where Muon has weaker performance compared to SGD. Namely, we find that Muon might struggle to uncover common underlying structure across tasks, and be more prone to fitting spurious features. By presenting these findings, we emphasize that the distinction between optimizers lies not only in convergence speed but also in their inherent inductive biases, which could be beneficial or not for any given problem.

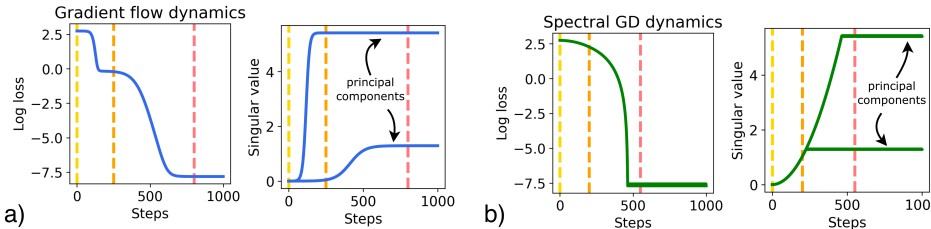

Figure 1: Illustration of the theory presented in Section B, for a) gradient flow and b) Spectral GD.

## 2 BACKGROUND AND THEORY ON DEEP LINEAR NETWORKS

**Muon.** Here we briefly recap optimization step of Muon (MomentUm Orthogonalized by Newton-Schulz), as introduced in Jordan et al. (2024b). Let $W^{(t)} \in \mathbb{R}^{d_{out} \times d_{in}}$ be a weight matrix after optimization step $t$, and $g^{(t)}$ the usual GD update of the weights with momentum, i.e. $g^{(t)} = \mu g^{(t-1)} + \nabla_W \mathcal{L}(W^{(t)})$. The update of Muon at time $t + 1$ aims to be $W^{(t+1)} = W^{(t)} - \eta U^{(t)} V^{(t)\top}$, where $g^{(t)} = U^{(t)} S^{(t)} V^{(t)\top}$ is the Singular Value Decomposition (SVD) of $g^{(t)}$[1]. However, computing the SVD at every update step is computationally prohibitive for deep learning, and Muon instead employs Newton-Schulz iterations as efficient approximation (Kovarik, 1970; Björck & Bowie, 1971).

**Spectral Gradient Descent.** We also introduce Spectral GD Carlson et al. (2015; 2016), as its optimization step is more tractable to analyze mathematically, but also keeps the important properties of Muon. Compared to Muon, Spectral GD uses precise SVD instead of Newton-Schulz iterations; it also doesn't use momentum, it orthogonalizes $\nabla_W L(W^{(t)})$ directly. Those are the only two differences, making analyzing Spectral GD relevant. We do so in the theoretical part where we think about the biases induced, but for our later experiments we switch back to using the full Muon.

**Deep Linear Networks: Spectral GD Loses Simplicity Bias of GD.** Given the implementational novelties Muon (or rather, Spectral GD for this paragraph) introduces compared to GD, we investigate how does this translate to their respective inductive biases. Here we present the informal theory and its consequences, and for full formality we refer to Appendix B.

**Theorem** (Gradient Flow Dynamics—Informal). *Gradient flow dynamics in 2 layer deep linear network exhibits gradual increase in complexity: starting from small initialization, principal components are learned one by one, from larger singular values $s$ to smaller, sequentially increasing the rank of the learned solution. For each principal component to be learned, a saddle point needs to be escaped, taking $\propto 1/s$ time.*

**Theorem** (Spectral Gradient Flow Dynamics—Informal). *In contrary, spectral gradient flow learns all principal components at the same time, with the same speed. Hence the ones with the smaller singular values $s$ are learned first, in time $\propto \sqrt{s}$. See Figure 1 for an illustration.*

**Consequences.** First, we see that geometrical reason GD is slower is because it has to break many saddle points along the way. This could motivate future optimizer development, if one is interested in traversing the loss landscape the same way GD does, while being faster. Spectral GD, on the other hand, doesn't pass through these (or other) saddles, but also it doesn't learn the solution gradually—it loses this type of simplicity bias. On the good side, Spectral GD can learn modalities appearing with different frequencies in the training set faster and more uniform than GD, as is pointed out by some prior work Vasudeva et al. (2025); Wang et al. (2025); Su (2025). On the other hand, such a learning is more *greedy*, in the sense that it rushes to learn all the different singular vectors quicker. Puli et al. (2023) showcases a setting where slower learning brings more benefits. In certain tasks simplicity bias is necessary, as it forms an implicit curriculum in learning.

From this theory, our intuition is that Muon would have a more greedy approach of memorizing, rather than discovering the underlying structure and generalizing. Guided by this, we came up with couple of experiments to compare Muon and SGD on. However, unlike most of (if not all) prior work, Muon performs worse than SGD due to losing the simplicity bias.

---

[1]This process of rescaling all non-zero singular values equal to 1 is termed *orthogonalization*.

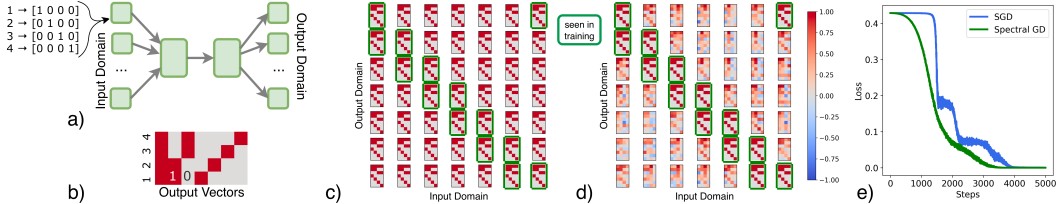

Figure 2: a) The neural network used to solve the task, where each of the gray arrows is a linear layer, with no nonlinearities in between. There is $M = 7$ input and output domains. Each input domain has its own, fixed 4 orthonormal vectors to represent $\{1, 2, 3, 4\}$. b) The underlying task we're learning: mapping each number in $\{1, 2, 3, 4\}$ to the output vector shown. Results after training in the 'routing' setup with c) SGD and d) Spectral GD, together with e) training loss curves. We plot the function the models learned (4 different column vectors represent the image of $\{1, 2, 3, 4\}$) for all the different input-output pairs of sources. Circled in green are the pairs seen during training.

## 3 SPECTRAL GD DOES NOT LEARN SHARED REPRESENTATIONS

First, we're testing Spectral GD in a task which can be solved in multiple ways, but only one—the one understanding the common underlying structure—is what we consider correct, as it achieves good out of distribution generalization. In that case, we say that the model has learned the 'shared representations'. Specifically, we use the 'routing' task setup from Saxe et al. (2022), a simplification of a multi-modal model. The architecture and task details are explained in Figure 2a,b). While all different input and output sources use their own linear encoders and decoders respectively, the common hidden layer is used by all. During training, each sample $(x, y)$ will have its input source (i.e. which domain $x$ came from) and output source (domain of $y$), and only corresponding layers are used and trained. Underlying task is very simple: the goal is to learn the mapping from 4 numbers $\{1, 2, 3, 4\}$ to 4 vectors, as shown in Figure 2b). However, each input source has a different vector representations for $\{1, 2, 3, 4\}$: for each input source, we generate four 4-dimensional orthonormal vectors, encoding the four numbers. For simplicity, we don't transform the output vectors based on output domain (but note that still, each output domain has their own linear decoder).

One could imagine this setting being a toy simplification for multi-modal networks, where perhaps different sources can include text, images, sound etc. Another example would be in healthcare, where data could be coming from different hospitals, and we'd like to develop a classification model that works for all input domains (e.g. different devices used to capture the data).

The catch of the task is that not all input-output pairs of sources are seen during training. Specifically, we only see samples $(x, y)$ where $x$ came from input source $j$, and $y$ is from either output domain $j$ or $j + 1$ ($M + 1 := 1$), for all $j = 1, ..., M$. The common linear layer is of dimension $64 \times 64$, thus large enough to memorize all of the mappings seen in training, even without realizing that the underlying task has a much simpler solution.

**Experiments.** The empirical results are captured in Figure 2c,d), where we show the function learned across all input-output pairs of sources. Training with SGD does result in learning the shared representations across all inputs, because even the input-output pairs that have never been seen during training do end up producing the correct underlying function. This generalization, however, doesn't happen when optimizing with Spectral GD, using stochastic setting with batches. While Spectral GD does learn to solve the task for input-output pairs seen during training, it does so by memorizing each input-output pair seen, as can be inferred from the poor generalization on input-output pairs not present in training. It is important to note that both algorithms achieve perfect convergence (Figure 2e), and the hope of Spectral GD to recovering the underlying shared representations with more data is gone. This example demonstrates that the removal of simplicity bias—while yielding faster convergence—imposes a tangible cost on the structural quality of the learned solution. By prioritizing rapid loss reduction, the optimizer bypasses the discovery of shared latent structures, converging instead to a disjointed, less generalizable solution. This failure to abstract shared rules poses a risk for complex tasks like mathematics, where the goal is not to memorize specific derivation steps, but to acquire reusable techniques applicable to unseen problems.

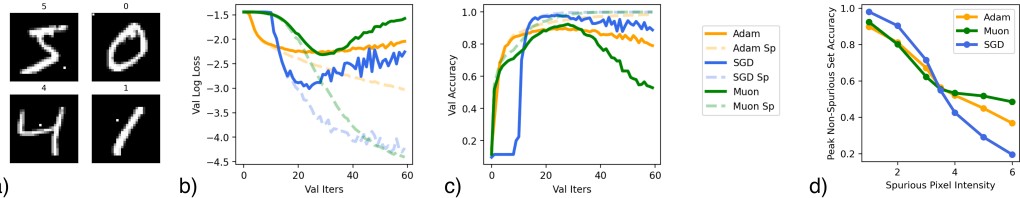

Figure 3: a) MNIST images containing spurious pixel. b) Validation losses and c) accuracies for SGD, Muon and Adam, on both sets with (Sp) and without spurious features. c) Peak accuracy on non-spurious dataset as a function of different intensities of the spurious pixel in the training data.

## 4    MUON VS SGD WITH SPURIOUS FEATURES

In the previous section, we focused on difference in the final solution SGD and Spectral GD found. Now, we focus on the differences in the trajectories. We use a setting with spurious correlations, where the data has some features we'd like the model to capture and some we don't. Unlike some setups with spurious correlations, here all training samples include the spurious features. Specifically, we use MNIST dataset, with the spurious feature being the intensity of a distinct pixel for each of the different classes. To solve the task, we train a simple CNN. As in our theory, and is usually done in practice, we initialize all the weights to be small (Gaussians). This time, we use Muon (with momentum and Newton-Schulz approximation), and compare it to both Adam and SGD with momentum. While we don't focus on Adam too much in this paper, here we note that, like Muon, Adam doesn't follow the GD trajectory, due to its per-parameter update rule Jacobs et al. (2026).

**Experiments.** The results are shown in Figure 3b,c). Two validation sets were used: one without spurious features and one with. We see that all three optimizers' performances started declining on the set without spurious features, meaning the solutions learned eventually started relying on the spurious pixel. Nonetheless, we see that the peak SGD accuracy on that set is better than the one for Adam or Muon. This means if one was to use early stopping as a regularization technique, SGD might be a better choice for an optimizer. More experimental details can be found in Appendix E.3.

From both accuracy and loss curves, we can see that SGD pursues a solution relying only on the actual digits for much longer—the Muon curves on with vs without spurious features sets start separating earlier. This somewhat relates to our theory: SGD learns the dominant principal component first, and only then starts learning the others; Muon (Spectral GD) learns all components in the same time. Analogously, learning the actual digits seems dominant in this setting, compared to relying on the spurious feature: and indeed, SGD first learns the dominant algorithm, and then starts learning the latter, where Muon seems to be learning both in the same time.

In Figure 3d), we vary the 'dominance' of relying on the actual digits by increasing the intensity of the spurious pixel. The dominance switch seems to happen at intensity $\approx 3.5$, where SGD becomes worse than others. Our theory explains this: once relying on the digit becomes less of a dominant solution, SGD's performance severely degrades as learning the digits starts only after the dominant spurious features have been almost fully learned. In contrast, Muon's accuracy hardly drops, because of the equal speed of learning of both digit and spurious features algorithms. More exploration in this direction seems promising, as it could help understand how different optimization algorithms find different functional solutions to a problem depending on the properties of the training data.

## 5    CONCLUSION

Motivated by the theory on deep linear networks, we aim to understand the biases of Muon optimizer, i.e. what are the trade-offs for its speed. We show its lack of simplicity bias does have consequences, e.g. not learning the shared representations. In a task with spurious features, we show how the theory translates to a non-linear setting, and observe that whether Muon or SGD is better depends on the data. Altogether, we hope this paper sparks new ideas in the field of deep learning optimizers, with higher emphasis on their biases and practical implications. Deciding on what makes a good optimizer is a hard question, and once we have an answer, it becomes easier to act upon it.

ACKNOWLEDGMENTS

The authors would like to thank Yedi Zhang and Jatin Prakash for useful discussions and feedback. This work was partly supported by the NIH/NHLBI Award R01HL148248, NSF Award 1922658 NRT-HDR: FUTURE Foundations, Translation, and Responsibility for Data Science, NSF CAREER Award 2145542, ONR N00014-23-1-2634, NIH R01CA296388, NSF 2404476, Optum, and Apple. This work was also supported by IITP with a grant funded by the MSIT of the Republic of Korea in connection with the Global AI Frontier Lab International Collaborative Research.

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

## A  RELATED WORK

We mostly base our theory in Section B for GD on Zhang et al. (2025a) and for Spectral GD on Vasudeva et al. (2025).

**Saddle-to-saddle dynamics and simplicity bias.**  Saddle-to-saddle dynamics Jacot et al. (2021); Abbe et al. (2023); Zhang et al. (2025a) is a phenomenon observed across architectures Maennel et al. (2018); Boix-Adsera et al. (2023); Zhang et al. (2025b), causing the learning to happen by gradually increasing the complexity of the solution. This gives rise to a particular type of simplicity bias Arpit et al. (2017); Gidel et al. (2019); Hu et al. (2020); Rahaman et al. (2019). As we as humans aim to find the simplest solutions possible (Occam's razor), it is important to us that models trained also have this intuition. That is why the question of simplicity bias in deep architectures has been broadly studied, both theoretically Valle-Perez et al. (2018) and empirically Nakkiran et al. (2019); Teney et al. (2022). Most of the theoretical papers used gradient descent for the optimizer Nakkiran et al. (2019); Gidel et al. (2019); Refinetti et al. (2023). This motivated us to think about the biases of other optimizers, especially Muon, as we can show it escape these saddles.

**Understanding Muon.**  From the theory side, Su (2025) analyses Muon optimizer on isotropic curvature model. Shen et al. (2025) provide a convergence rate analysis of (simplified), in comparison with GD.

Most of the empirical papers focus on the benefits of Muon. In Vasudeva et al. (2025) they compare optimizers on imbalanced classes setting using MNIST-CIFAR dataset and Tiny Stories for language generation. Wang et al. (2025) show that in a transformer, MLP and OV-circuit benefit the most from Muon (versus Adam), and also, motivated by associative memory model, show that Muon outperforms Adam on a memorization task(QA dataset about biographical information) task where each dataset entry appears with different frequency. They state '*Muon consistently yields more isotropic weight matrices with broadly distributed spectral energy than Adam, both throughout training and across random initializations, thereby supporting richer feature representations*'. In Section 3, we gave an example where 'richer' in their sense (i.e. learning everything there is to learn) is actually bad. Both of these works emphasize how Muon is better in the setting of imbalanced data, learning all modalities more evenly.

Vasudeva et al. (2025) also provides a comparison of the optimizers in a setting with spurious correlations, however, different than ours. They use data where a percentage (99%) of samples contains the spurious feature (color in MNIST). While they show all optimizers compared do learn the task, they emphasize that Muon learns it the fastest. They setting also differs because they say color (the spurious feature) is more dominant than digit shape, which perhaps explains why SGD learns it only later. In our setting, we hypothesize that shape is more dominant than the spurious pixel, which could completely change the conclusion. They nonetheless state: '*This suggests that Muon's spectral design promotes balanced learning of these components, leading to superior generalization.*'. We showed a setting where Muon actually doesn't generalize but SGD does!

Most of the prior work focuses on benefits of Muon, rarely (we're not aware of any work) providing a setting where Muon is worse. We believe it's important to showcase Muon's failures as well, first to prevent people over-claiming and glorifying Muon, and also to serve as a motivation for future improvements.

## B  THEORY

In this work, we build on the theory explaining a simplicity bias of GD Nakkiran et al. (2019); Gidel et al. (2019); Zhang et al. (2025a). We use the well-studied framework of deep linear networks Saxe et al. (2013; 2019); Lampinen & Ganguli (2018); Ji & Telgarsky (2019); Jacot et al. (2021); Ziyin et al. (2022), particularly interesting for understanding the learning dynamics in deep models. Building on prior work, which usually assumes the optimization to be performed by gradient descent, we theoretically analyze Spectral GD optimization dynamics. Namely, under mild assumptions, both optimization algorithms lead to a mathematically tractable systems. While GD gradually increases the complexity of the solution, Muon doesn't have this property.

In this section, we focus on 2 layer deep linear networks. First, we lay down the theory behind the learning dynamics with GD Saxe et al. (2013); Gidel et al. (2019); Jacot et al. (2021); Zhang et al. (2025a), and Spectral GD Vasudeva et al. (2025). Once we have the theory, we comment on the consequences—Spectral GD and GD traverse different learning trajectories in the loss landscape. The theory explains why Spectral GD is learning faster, but also reveals the cost of it: loss of the simplicity bias of GD.

Let a forward pass of a 2 layer deep linear network be $\hat{y} = VUx$, where $U \in \mathbb{R}^{H \times d_{\text{in}}}, V \in \mathbb{R}^{d_{\text{out}} \times H}$, $x \in \mathbb{R}^{d_{\text{in}}}$, and $H$ is the number of hidden neurons. Let the rows of $U$ be $u_i^{\top}$, the columns of $V, v_i$. We are given $n$ input-output pairs $\{(x_i, y_i)\}$. The goal is to minimize the MSE loss $L(U, V) = \frac{1}{2n} \sum_i \|\hat{y}_i - y_i\|^2$. Here we use gradient flow on the whole dataset. Dynamics dependence on the dataset is fully captured by these statistics: $\Sigma_{xx} = \frac{1}{n} \sum_{i=1}^{H} x_i x_i^{\top}$ and $\Sigma_{yx} = \frac{1}{n} \sum_{i=1}^{H} y_i x_i^{\top}$.

**Assumptions:** For the clarity of presentation, we assume $\Sigma_{xx} = I$, and for more generality we refer to Zhang et al. (2025a); Vasudeva et al. (2025); Watanabe et al. (2026). We also assume the joint diagonalizability of $\Sigma_{yx}, U_0$ and $V_0$, which are the weight matrices at initialization. This is a reasonable assumption, as we'll always start from small initialization and prior work Zhang et al. (2025a) shows that in that regime, the principal components of $U, V$ do align with the ones of $\Sigma_{yx}$ early on in the training—a phenomena known as silent alignment Hu et al. (2020); Jacot et al. (2021).

**Gradient flow solution.** The gradients of $L$ with respect to $U$ and $W$ are given by

$$\nabla_U L = V^{\top}(VU\Sigma_{xx} - \Sigma_{yx}), \nabla_V L = (VU\Sigma_{xx} - \Sigma_{yx})U^{\top}.$$

Let $\Sigma_{yx} = \sum_{k=1}^{D} s_k q_k r_k^{\top}$ be the SVD of $\Sigma_{yx}$ where $s_k$ are positive singular values. Let them be ordered $s_1 \geq ... \geq s_D$.

**Theorem B.1** (Gradient Flow Dynamics). *Consider the gradient flow dynamics $\dot{U} = -\nabla_U L$ and $\dot{V} = -\nabla_V L$ on the loss $L(U, V)$ starting from infinitesimal initialization. Then:*

1. *Critical Points: Any point $(U, V)$ such that $VU = \sum_{k=1}^{r} s_k q_k r_k^{\top}$ (for $r \leq D$) with $u_i \in span\{r_k\}_{k=1}^{r}$, $v_i \in span\{q_k\}_{k=1}^{r}$ is a critical point of the dynamics.*

2. *Sequential Learning: The $k$-th singular value $\sigma_k(t)$ of the product matrix $VU$ evolves according to a sigmoid function, with a convergence time $\propto s_k^{-1}$. Consequently, for any $k$ such that $s_k > s_{k+1}$, there exists a time interval when the $k$-th mode is fully learned ($\sigma_k \approx s_k$) while the $(k+1)$-th mode is negligible ($\sigma_{k+1} \approx 0$).*

The proof, although standard, can be found in Appendix C. Combining the two results leads to the explanation of saddle-to-saddle dynamics from small initialization. At the start of training, all singular values of the solution $W$ are effectively 0. Rank of $W$ then increases gradually, first learning the highest singular values $s_k$. At the point rank has gone up to $r$ (for all $r \leq D$), $VU$ effectively satisfies the conditions of 1, and the learning trajectory passes close to the corresponding saddle point. This is reflected in a loss plateau, and 2. predicts the time of escaping this saddle. In case of a singular value $s_k$ having multiplicity $m > 1$, the rank at that point will increase by $m$ and the whole singular subspace will be learned at the same time.

**Spectral gradient descent solution.** Here we use the same notation and assumptions as in the previous case. For spectral gradient descent, the dynamics is different. This time, all principal components are learned simultaneously, with the same speed, until each one saturates.

**Theorem B.2** (Spectral Gradient Flow Dynamics). *Observe the spectral gradient flow $\dot{U} = -orth(\nabla_U L)$ and $\dot{V} = -orth(\nabla_V L)$ where $orth(\cdot)$ is the orthogonalization of a matrix. Starting from infinitesimal initialization, we have:*

1. *Solution Trajectory: The learning trajectory of $W = VU$ will sequentially pass through all $W_r := \sum_{k=1}^{r} s_{r+1} q_k r_k^{\top} + \sum_{k=r+1}^{D} s_k q_k r_k^{\top}$ for $r \leq D$ (from higher to lower) such that $s_r > s_{r+1}$.*

2. *Equal Learning: To get from one $W_r$ to the next, $W$ learns singular spaces of $\Sigma_{yx}$ in the same time, at the same rate, until saturation. Furthermore, the evolution of $k$-th singular value $\sigma_k(t)$ of $W$ follows a quadratic curve, leading to convergence time $\propto \sqrt{s_k}$.*

We provide the proof in Appendix D, or refer to Vasudeva et al. (2025), with slightly different assumptions. Consequently, training with Spectral GD also happens in phases: in each phase, singular values $\sigma_{r+1}(t) = s_{r+1}, ..., \sigma_D(t) = s_D$ are fully learned, while $\sigma_1(t), ..., \sigma_r(t)$ make their transition from $s_{r+1}$ to $s_r$, the next smallest singular value. The contrast with GD is the order of learning the singular values: while GD fully learns the higher ones first, Spectral GD learns all of them in the same time, leading to smaller ones being fully learned first.

**Experiments.** We both validate and illustrate the theory in Figure 1, where we perform full dataset (Spectral) GD on standard Gaussian data $x_i \in \mathbb{R}^{d_{\text{in}}}$, and $y_i \in \mathbb{R}^{d_{\text{out}}}$ computed as a noised linear function of $x_i$. In this usual linear regression setting, we sample the regression weights also as a standard normal. The weights $U, V$ are initialized also as Gaussian, but with small variance (0.01). We see that the evolution closely follows the theory. More details on the figure and experimental setup can be found in Appendix E.1.

Empirically, because we're performing a discretization of the dynamics of 'Spectral gradient flow', and due to using a spectral method, it does happen that after the last $D-r$ principal components have been learned, the update matrix in the dynamics above is not exactly rank $r$ but also have some noise, and is usually full rank. In that case, Spectral GD does fit this noise, causing a highly oscillatory path in the loss landscape (see Figure 5). Although in a simple setting of deep linear networks the model always converges to the correct solution (as it is a stable minimum), this suggest that in more complicated settings Spectral GD and Muon may be less robust when using higher learning rate. Potentially, this observation could motivate future improvements of spectral methods.

## C PROOF OF THEOREM B.1

*Proof.* **Critical Points.** The gradients for the deep linear network under $\Sigma_{xx} = I$ are:

$$\nabla_U L = V^\top(VU - \Sigma_{yx}),$$
$$\nabla_V L = (VU - \Sigma_{yx})U^\top.$$

Let the candidate solution be $W_r = VU = \sum_{k=1}^r s_k q_k r_k^\top$. Substituting this into the residual term $(VU - \Sigma_{yx})$ yields:

$$W_r - \Sigma_{yx} = \sum_{k=1}^r s_k q_k r_k^\top - \sum_{k=1}^D s_k q_k r_k^\top = -\sum_{k=r+1}^D s_k q_k r_k^\top.$$

By the theorem assumption, $\text{row}(U) \in \text{span}\{r_k\}_{k=1}^r$ and $\text{col}(V) \in \text{span}\{q_k\}_{k=1}^r$, hence:

$$\nabla_U L = -V^\top \left( \sum_{k=r+1}^D s_k q_k r_k^\top \right) = 0,$$

$$\nabla_V L = -\left( \sum_{k=r+1}^D s_k q_k r_k^\top \right) U^\top = 0.$$

Thus, $(U, V)$ is a critical point.

**Sequential Learning.** This is a standard result found in many deep linear networks work Saxe et al. (2013); Gidel et al. (2019); Li et al. (2020); Jacot et al. (2021); Zhang et al. (2025a). For completeness, we provide a proof here. We analyze the evolution of the product matrix $W = VU$. We assume the standard "balanced" condition $V^\top V = UU^\top$, which is an invariant/conservation law of gradient flow if initialized infinitesimally, as we assume. The dynamics of the product matrix $W$ are given by:

$$\dot{W} = \dot{V}U + V\dot{U} = -(W - \Sigma_{yx})U^\top U - VV^\top(W - \Sigma_{yx}). \tag{1}$$

Under the balanced condition, we have $U^\top U = (W^\top W)^{1/2}$ and $VV^\top = (WW^\top)^{1/2}$. Substituting this into the dynamics yields:

$$\dot{W} = -(W - \Sigma_{yx})(W^\top W)^{1/2} - (WW^\top)^{1/2}(W - \Sigma_{yx}). \tag{2}$$

$W(0)$ and $\Sigma_{yx}$ are jointly diagonalizable in the SVD basis of $\Sigma_{yx}$, which holds by assumption. Since the gradient update is a function solely of $W$ and $\Sigma_{yx}$, this alignment is preserved throughout the flow.

Let $W(t) = \sum_{k=1}^{D} \sigma_k(t) q_k r_k^\top$. Substituting this spectral decomposition into the matrix ODE decouples the system into independent scalar differential equations for each singular value $\sigma_k(t)$:

$$\dot{\sigma}_k(t) = -(\sigma_k - s_k)\sigma_k - \sigma_k(\sigma_k - s_k) = 2\sigma_k(t)(s_k - \sigma_k(t)). \tag{3}$$

This is the logistic differential equation. The solution is:

$$\sigma_k(t) = \frac{s_k}{1 + \left(\frac{s_k}{\sigma_k(0)} - 1\right) e^{-2s_k t}}. \tag{4}$$

This solution shows that $\sigma_k(t)$ follows a sigmoidal trajectory, saturating to $s_k$. Consequently, the time taken for $s_k$ to be learned is of order $1/s_k$, implying that modes with larger singular values are learned significantly faster, establishing the sequential learning property. $\qquad\square$

## D  PROOF OF THEOREM B.2

*Proof.* **Solution Trajectory.** The proof, with slightly different assumptions, can be found in Vasudeva et al. (2025). Nonetheless, we follow our setup and provide it here for completeness.

We assumed joint diagonalizibility of $U(0) = \sum_{k=1}^{D} \sigma_k^U(0) z_k r_k^\top$, $V(0) = \sum_{k=1}^{D} \sigma_k^V(0) q_k z_k^\top$ and $\Sigma_{yx} = \sum_{k=1}^{D} s_k q_k r_k^\top$, which is justified by the infinitesimal initialization. As we'll see, once $U(t), V(t)$ are jointly diagonalizable in this way, they stay so for all $t' \geq t$. Suppose for some $r \leq D, t_0 \geq 0$, $W(t_0) = \sum_{k=1}^{r} s_{r+1} q_k r_k^\top + \sum_{k=r+1}^{D} s_k q_k r_k^\top$. Indeed, this is satisfied at initialization $t_0 = 0$, with $r = D$, once we denote $s_{D+1} = 0$. If it holds at $t_0$, then

$$\nabla_U L(U(t_0)) = V(t_0)^\top (W(t) - \Sigma_{yx})$$

$$= (\sum_{k=1}^{D} \sigma_k^V(t_0) q_k z_k^\top)^\top (\sum_{k=1}^{r} (\sigma_{r+1} - \sigma_k) q_k r_k^\top)$$

$$= \sum_{k=1}^{r} \sigma_k^V(t_0)(\sigma_{r+1} - \sigma_k) z_k r_k^\top$$

$$\nabla_V L(V(t_0)) = (W(t_0) - \Sigma_{yx}) U(t_0)^\top$$

$$= (\sum_{k=1}^{r} (\sigma_{r+1} - \sigma_k) q_k r_k^\top)(\sum_{k=1}^{D} \sigma_k^U(t_0) z_k r_k^\top)^\top$$

$$= \sum_{k=1}^{r} \sigma_k^U(t_0)(\sigma_{r+1} - \sigma_k) q_k z_k^\top$$

From this, the orthogonalizations of the gradients are $\mathrm{orth}(\nabla_U L(t_0)) = -\sum_{k=1}^{r} z_k r_k^\top$, $\mathrm{orth}(\nabla_V L(t_0)) = -\sum_{k=1}^{r} q_k z_k^\top$. This means that change in $U(t), V(t)$ happens only in the first $r$ singular vectors, and the dynamics will look as a gradient flow $\dot{U}(t) = \sum_{i=1}^{r} z_k r_k^\top$, $\dot{V}(t) = \sum_{k=1}^{r} q_k z_k^\top$. This keeps the matrices jointly diagonalizable, and therefore we can observe the dynamics on the singular values, from initial conditions at $t_0$:

$$\dot{\sigma}_k^V(t) = \mathbb{1}_{\{\sigma_k^V(t)\sigma_k^U(t) < s_k\}}$$

$$\dot{\sigma}_k^U(t) = \mathbb{1}_{\{\sigma_k^V(t)\sigma_k^U(t) < s_k\}},$$

noting that $\sigma_k(t) = \sigma_k^V(t)\sigma_k^U(t)$. By symmetry at infinitesimal initialization, all $\sigma_k^V(t)$ and $\sigma_k^U(t)$ will be the same for $k \leq r$, and hence for all $k \leq r$, $\sigma_k(t)$ will evolve the same, from $\sigma_k(t_0) = s_{r+1}$ to $\sigma_k(t_1) = s_r$. At time $t_1$, for all $m < k \leq r$ with $m$ largest s.t. $s_{m+1} = s_r$, the growth of $\sigma_k(t)$ stops. Hence the solution $W$ passes through $W_m$.

**Equal learning.** Continuing with the notation above, if we start from $\sigma_k^V(0) = \sigma_k^U(0) = 0$, the solution of the system is $\sigma_k^U(t) = \sigma_k^U(t) = \min(t, \sqrt{s_k})$, and hence $\sigma_k(t) = \min(t^2, s_k)$. This shows

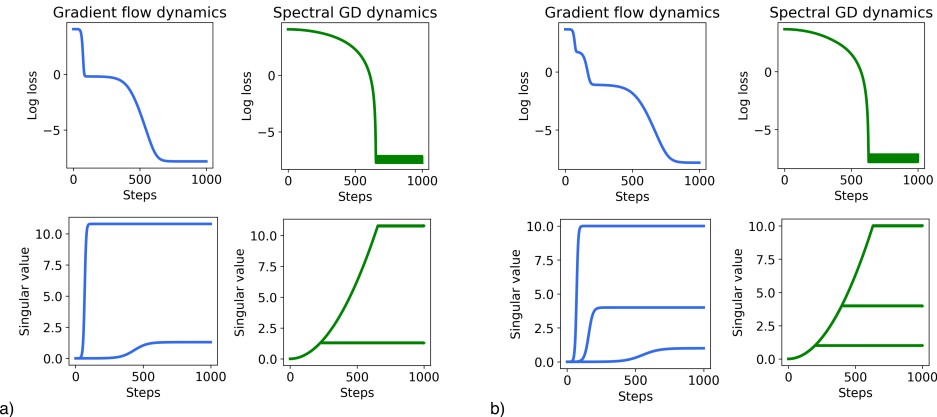

Figure 4: Additional figures supporting the theory from Section B. a) $d_{\text{in}} = d_{\text{out}} = 2$; b) $d_{\text{in}} = d_{\text{out}} = 3$

both the equal growth and the quadratic curve for learning singular values of $\Sigma_{yx}$. Consequently, the time to learn $\sigma_k(t) = s_k$ is $t \propto \sqrt{s_k}$.

$\square$

## E    FURTHER EXPERIMENTS AND DETAILS

### E.1    DEEP LINEAR NETWORKS

With the notation introduced in Appendix B, we provide details on Figure 1. On the left is the loss curve, and on the right the evolution of singular values of $VU$. Here $d_{in} = d_{out} = 2$ and $H = 100$. We observe that for GD, first singular vector is fully learned first (Step 250, orange line), and only then the second one is learned (Step 800, red line). On the other hand, Spectral GD learns both of them in the same time, and after it saturates on the smallest one (Step 200, orange line), then $u_i$s progress only in the direction of the larger one (Step 550, red line). Simulation is closely following the theory, as expected. Some more analogous experiments can be found in Figure 4.

Additionally, we show the oscillations of the singular values of $W$ during training, happenning as a consequence of our discussion in Section B.

### E.2    NEURAL RACE REDUCTION AND SHARED REPRESENTATIONS

**Analysis.**    In a similar setup, Saxe et al. (2022) theoretically demonstrate that GD has an implicit bias towards learning shared representations, with the core argument lying in Theorem B.1 2. Because of our gained intuition, we identified this setting as the one where Spectral GD may be disadvantaged by the absence of simplicity bias. Mathematically, each gating mechanism follows the dynamics derived by Saxe et al. (2022):

$$\frac{d}{dt}B_1 = \frac{\sqrt{P}}{M^2}B_2B_1\left[S - B_2B_1^2D\right]$$
$$\frac{d}{dt}B_2 = \frac{P}{M^2}B_1^2\left[S - B_2B_1^2D\right]$$

Here, $B_1$ and $B_2$ represent the spectra/mode strengths of the gated components, while $S$ and $D$ denote certain data statistics (see Saxe et al. (2022)). The focus is on "pathway counting" argument: the gradient updates are scaled by pre-factors proportional to $P$, the number of pathways passing through a gated unit. In the initial phase of learning, where dynamics are dominated by exponential growth, these pre-factors act as rate constants; consequently, configurations with maximal $P$ (i.e., $P = M^2$, corresponding to shared representations) are exponentially faster to learn and win the "neural race". In contrast, Spectral GD orthogonalizes each of the updates, effectively removing the dependence of learning speed on the pathway multiplicity $P$, and the order of speed of learning for

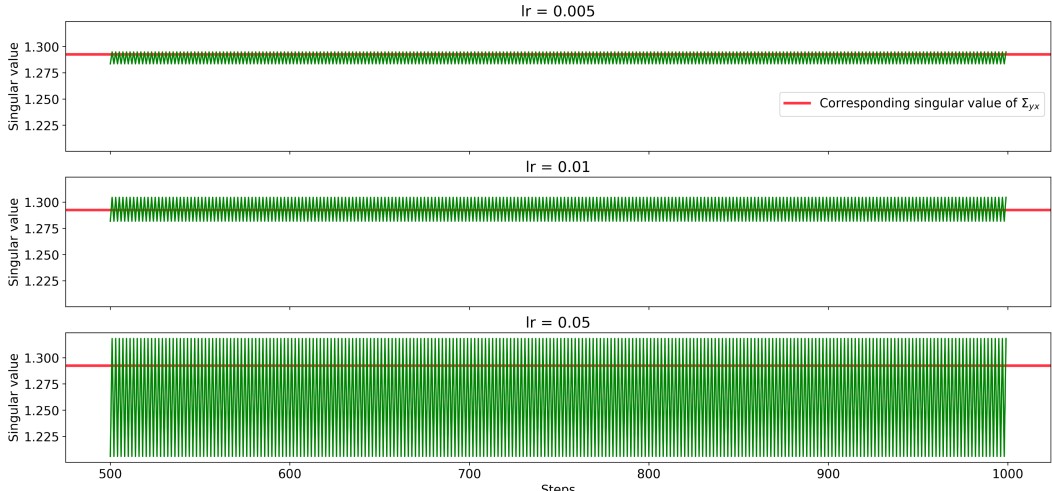

Figure 5: The oscillations of $\sigma_2(t)$ around $s_2$ in the setting from Figure 1, shown for different values of the learning rate. This phenomena happens after a principal component is effectively learned by Spectral GD, but not exactly. Then the small noise in the direction of that principal component is amplified by orthogonalization, and the step of order 1 is taken, independently of noise magnitude.

each gating mechanism is the same. In that case, the winning gating strategy is dependent on other conditions, such as the random initialization.

We verified this intuition in our experiments in Figure 2. Further confirmation comes from the hidden layer's spectral properties: whereas the SGD solution converges to an approximate low-rank structure (rank = 4, equaling the dimension of the underlying task), the Spectral GD solution exhibits a significantly higher effective rank with a heavy-tailed spectrum—a clear signature of memorization rather than structural learning.

### E.3 SPURIOUS FEATURES

We inject a single pixel artifact at a fixed, class-specific location in every training image, offering the model a simple heuristic to classify digits.

Here we provide more implementational details on the results in Figure 3. For all three optimizers in b,c,d), we picked the best performing hyperparameters across multiple runs. We used Muon implementation from Jordan et al. (2024b), where we did experiment with different learning rates for parameters optimized by Muon versus Adam. For Figure 3b,c), spurious pixel intensity was set to 1. We used a simple CNN with 2 convolutional layers followed by 2 linear layers. Weights are initialized as Gaussian, with mean 0 and standard deviation 0.01. Convolution kernel sizes are 3, and the number of filters is 32 in the first and 64 in the second layer, with MaxPool of size 2 in between. The first fully connected layer has output dimension 128, and the second one 10 (the number of classes).

