# OpenReview forum: "To Use or not to Use Muon: How Simplicity Bias in Optimizers Matters"
_ICLR.cc/2026/Workshop/Sci4DL — Sci4DL 2026_

### Official Review · Reviewer_cWgX · 2026-02-10

**Fit:** 3
**Significance:** 2
**Confidence:** 2

**Summary:**

In this paper, the authors explored the downsides of training with the Muon optimizer. Specifically, the authors argued that Muon optimizers provide a noticeable speedup in training while losing simplicity bias. Through theoretical analysis, the authors also showed that while gradient descent learns solutions gradually, Spectral GD learns all the principal components at once. A routing task and a spurious feature MNIST examples presented in this paper demonstrate that Muon can fail to learn shared representations and rely on spurious features during training.

**Strengths:**

While Muon optimizer isn’t as popular as other optimizers such as Adam or SGD, it has gained tractions in recent years by its lower training time compared to others. The authors provided new insights into the disadvantages and inductive  biases of using Muon optimizer. The authors framed the discussion around simplicity bias gives a unifying scope for theoretical analysis and experiments.

**Suggestions:**

It is understood that Spectral GD is used in the routing task for theoretical analysis and clarity. For completeness, it’s also interesting to see if the same failure mode persists when using full Muon. This is an interesting study, and natural future direction could potentially include validation in other models such as a small transformer. And it would also be valuable to derive practical such as when Muon is likely to be beneficial or not.

---

### Official Review · Reviewer_Vo4D · 2026-03-01

**Fit:** 3
**Significance:** 2
**Confidence:** 2

**Summary:**

This paper studies optimizer-induced inductive bias, focusing on Muon. The main claim is that Muon (modeled analytically via Spectral GD / spectral gradient flow) trades off GD/SGD’s simplicity bias for faster, more “uniform” learning across modes. In 2-layer deep linear networks, the paper argues that gradient flow learns principal components sequentially (gradual rank increase), while spectral methods learn components more simultaneously, which can explain speedups but also suggests different solution preferences. Guided by this, the authors present (i) a controlled routing / multi-domain toy task where SGD recovers a shared latent rule and generalizes to unseen domain pairs, while Spectral GD fits the training pairs but fails to learn shared representations; and (ii) an MNIST setting with a class-specific spurious pixel where Muon (and Adam) latch onto the spurious cue earlier than SGD, affecting peak accuracy and highlighting trajectory/early-stopping effects.

**Strengths:**

1. Good workshop fit. Clear “science of DL” approach: minimal theory + targeted experiments to test hypotheses about training dynamics and inductive bias (not SOTA chasing).
2. Timely and important question. Muon is being adopted widely; understanding when its bias helps or hurts is valuable.
3. Concrete “failure mode” examples. The routing task nicely illustrates that low training loss can correspond to qualitatively different solutions (shared rule vs memorization) depending on optimizer.
4. Trajectory emphasis. The spurious-feature experiment usefully emphasizes when an optimizer starts exploiting shortcuts, connecting naturally to early stopping and implicit curriculum.

**Suggestions:**

1. **Tighten the Muon - Spectral GD link (and avoid overclaiming).**
   The theory is for Spectral GD (exact SVD, no momentum), while Muon uses Newton–Schulz approximations and usually momentum; in practice Muon may also be applied only to certain parameter blocks. The routing experiment is shown with Spectral GD, not Muon, which makes the "Muon may fail to learn shared reps" conclusion feel a bit indirect.
   * Suggestion: run the routing/shared-representation experiment with actual Muon (and ideally an ablation: Muon w/ vs w/o momentum; or varying Newton–Schulz iterations) to confirm the phenomenon transfers.
2. **Add stronger baselines in the routing task.**
   The spurious MNIST section includes Adam (good), but the routing results would be much more convincing if we saw whether this is "Spectral/Muon-like methods" broadly or something specific.
   * Suggestion: include Adam (and optionally signSGD+momentum as simplified version) on the routing setup.
3. **Show learning curves for the *right* metric, not just final snapshots.**
   The routing section argues Spectral GD's hope of recovering shared reps "is gone," but it's hard to rule out transient structure that early stopping might capture.
   * Suggestion: plot generalization to unseen domain pairs vs training step, alongside loss. This would also make the "simplicity-bias-as-implicit-curriculum" story more compelling and test whether early stopping rescues Muon/Spectral GD.
4. **Regularization/capacity controls.**
   In routing, the shared layer is explicitly large enough to memorize. That's fine for the point being made, but it raises the question of whether the gap is really "optimizer bias" vs "capacity \+ lack of regularization."
   * Suggestion: add ablations on hidden size, weight decay, and/or other regularization. If the effect persists across these, the claim becomes much stronger.
5. **External validity: one more "bridge" experiment or sharper scoping.**
   The deep-linear theory is a good minimal model, and MNIST is a real dataset, but the practical implications for modern training (transformers/diffusion models) remain somewhat speculative.
   * Suggestion: either add a small-scale modern experiment (even a tiny transformer in a controlled multi-task/shared-structure setting), or explicitly scope the claims as *hypotheses* suggested by the minimal model \+ toy evidence.
6. **Writing / presentation small issues.**
   * The phrase "morally correct" reads informal/loaded; I'd suggest replacing it.
   * Citation links appear broken in the PDF.
   * Consider plotting the routing loss on a log scale; it may reveal phases/plateaus more clearly and better match the simplicity-bias narrative.

---

### Meta-Review · Area_Chair_Xn2k · 2026-03-01

**Recommendation:** Accept

**Metareview:**

The work discusses the strengths and limitations of muon optimizer. Particularly the biases that come with muon and the consequences of the learning trajectories that a model takes. I think this work is interesting and I recommend and accept.

---

### Decision · Program_Chairs · 2026-03-02

Accept